# Commercial airline protocol during COVID-19 pandemic: An experience of Thai Airways International

Krit Pongpirul[1,2,3]*, Kanitha Kaewpoungngam[4], Korn Chotirosniramit[1], Sinnop Theprugsa[5]

1 Department of Preventive and Social Medicine, Faculty of Medicine, Chulalongkorn University, Bangkok, Thailand, 2 Department of International Health, Johns Hopkins Bloomberg School of Public Health, Baltimore, MD, United States of America, 3 Bumrungrad International Hospital, Bangkok, Thailand, 4 College of Aviation Development and Training, Dhurakij Pundit University, Bangkok, Thailand, 5 Thai Airways International Public Company Limited, Bangkok, Thailand

* doctorkrit@gmail.com

## Abstract

### Introduction

Coronavirus disease 2019 (COVID-19) pandemic has affected the aviation industry. Existing protocols have relied on scientifically questionable evidence and might not lead to the optimal balance between public health safety and airlines' financial viability.

### Objective

To explore the implementation feasibility of Thai Airways International protocol from the perspectives of passengers and aircrews.

### Design

An online questionnaire survey of passengers and an in-depth interview with aircrews.

### Setting

Two randomly selected repatriation flights operated by Thai Airways International using Boeing 777 aircraft (TG476 from Sydney and TG492 from Auckland to Bangkok)

### Participants

377 Thai passengers and 35 aircrews.

### Results

The mean age of passengers was 28.14 (95%CI 26.72 to 29.55) years old; 57.03% were female. TG492 passengers were mostly students and significantly younger than that of TG476 (p<0.0001) with comparable flying experience (p = 0.1192). The average body temperature was 36.52 (95%CI 36.48 to 36.55) degrees Celsius. Passengers estimated average physical distances of 1.59 (95%CI 1.48 to 1.70), 1.41 (95%CI 1.29 to 1.53), and 1.26

**Data Availability Statement:** All relevant data are within the paper and its Supporting Information files. Data are also available from the UK Data Service at https://beta.ukdataservice.ac.uk/

myaccount/deposits/datasets/under-deposit?id=
4b7b1b10-d57f-407d-a95f-11125749fd39#!/
catalogue.

**Funding:** Thai Airways International provided
support in the form of salaries for author ST, but
did not have any additional role in the study design,
data collection and analysis, decision to publish, or
preparation of the manuscript. The specific roles of
the authors are articulated in the 'author
contributions' section.

**Competing interests:** The authors declare that they
have no conflicts of interest. The commercial airline
affiliation of author ST does not alter our adherence
to PLOS ONE policies on sharing data and
materials.

(95%CI 1.12 to 1.41) meters at check-in, boarding, and in-flight, respectively. Passengers were checked for body temperature during the flight 1.97 (95%CI 1.77 to 2.18) times on average which is significantly more frequent in longer than shorter flight (p<0.0001). Passengers moved around or went to the toilet during the flight 2.00 (95%CI 1.63 to 2.37) and 2.08 (95%CI 1.73 to 2.43) times which are significantly more frequent in longer than shorter flight (p = 0.0186 and 0.0049, respectively). The aircrews were satisfied with the protocol and provided several practical suggestions.

## Conclusion

The protocol was well received by the passengers and aircrews of the repatriation flights with some suggestions for improvement.

## Introduction

Coronavirus disease 2019 (COVID-19) pandemic has affected several industries including aviation. Transmission of the severe acute respiratory syndrome (SARS) associated coronavirus (SARS-CoV) on aircraft was reported—individual with physical proximity to the index symptomatic patient (three rows in front) have approximately three times the risk of the passengers who seated elsewhere [1]. Despite many similarities with SARS-CoV, the novel coronavirus (SARS-CoV-2) appears to transmit more easily than its predecessor. A recent study reported potential transmission from asymptomatic COVID-19 infected individuals [2], suggesting that symptom-based case detection might be no longer adequate [3]. A commercial airline has begun carrying out serology tests on passengers before departure [4] in addition to temperature screening.

Given no specific and robust evidence on the risk of in-flight transmission of the SARS-CoV-2, preventive measures relied on the past experiences; at least 275 options have been proposed to reduce SARS-CoV-2 transmission in five key areas: (1) physical isolation, (2) reducing transmission through contaminated items, (3) enhancing cleaning and hygiene, (4) reducing spread through pets, and (5) restricting disease spread between areas [5].

While several preventive activities have been agreed upon by stakeholders, some measures have raised financial and feasibility concerns to the airline industry. An optimal balance between public health safety and airline financial viability is critical, especially when the airline passenger revenues already dropped by $314 billion in 2020 [6]. General biosecurity measures such as temperature screening of individuals, minimizing inter-personal contacts during the boarding and deplaning processes, limiting movement within the cabin during flight, increasing frequency and quality of cabin cleaning, and simplifying catering procedures [6] have been implemented at the expense of the aviation industry. The International Air Transport Association (IATA) recently endorsed the mandatory face-coverings for passengers and masks for crew members but opposes onboard social distancing because of the significant loss of revenue [6]. IATA asserted that the risk of infectious disease transmission on board is low even without special measures as suggested by scientifically questionable evidence such as contact tracing for selected flights or informal surveys of major airlines [6]. However, proving the effectiveness of these multi-faceted measures have been difficult. Also, these might not be well perceived or complied by some passengers.

Like other national aviation authorities, the Civil Aviation Authority of Thailand (CAAT) has issued a temporary ban on all international flights to Thailand during the COVID-19

**Table 1. COVID-19 risk score, The Civil Aviation Authority of Thailand (CAAT).**

| Score | 1 | 2 | 3 | 4 | 5 |
|---|---|---|---|---|---|
| Number of Covid-19 Cases in Country of Departure | < 50 | 50–100 | 101–500 | 501–1,000 | >1,000 |
| Proportion of Seats Occupied with Passengers (%) | < 40 | 40–80 | > 80 | - | - |
| Flight Duration (hours) | < 4 | 4–8 | > 8 | - | - |
| Risk-based Interventions: | | | | | |
| Low Risk (score 3–4) | Passengers: Body temperature check by using a non-contact infrared thermometer before boarding. Passengers with body temperature higher than 37.3 degree Celsius or upper respiratory tract symptoms (cough, sore throat, running nose, and shortness of breath) will be reassessed by Port Health Officer if a boarding pass could be given. | | | | |
| | Crews: Disposable medical or surgical masks. | | | | |
| | Pilots: Disposable medical or surgical masks. | | | | |
| Moderate Risk (score 5–7) | Passengers: Body temperature check by using a non-contact infrared thermometer before boarding and in-flight for long-haul (>4 hours) flights. | | | | |
| | Crews: Disposable medical or surgical masks. | | | | |
| | Pilots: Disposable medical or surgical masks. | | | | |
| High Risk (score 8–11 or no HEPA[a] filtering system) | Passengers: Body temperature check by using a non-contact infrared thermometer before boarding and in-flight for long-haul (>4 hours) flights. | | | | |
| | Crew: N95 or surgical masks, goggles, and disposable rubber gloves. | | | | |
| | Pilots: Surgical masks and goggles. | | | | |

[a]High-Efficiency Particulate Air

pandemic with some exceptions [7]. Several commercial airlines, including Thai Airways International, have been able to operate repatriation flights, organized in coordination with governments to aid citizens stranded abroad. Individuals must fill in and submit the Application for Re-entry Permit to Return into the Kingdom (TM.8) to an immigration officer [8] and the COVID-19 Screening Questionnaire (T.8) to Port Health Officer [9]. The COVID-19 risk score is calculated by using three factors: the number of COVID-19 case in the country of departure, the proportion of seats occupied by the passengers, and flight duration into low, moderate, and high risks (Table 1). Flight without the High-Efficiency Particulate Air (HEPA) filtering system is considered high risk.

These repatriation flights offer a wonderful opportunity to gather useful evidence, especially from the passengers' perspective, for commercial airline protocol development and improvement. This study aimed to explore the implementation feasibility of the Thai Airways International protocol from the perspectives of passengers and aircrews.

## Methods

We conducted an online questionnaire survey of passengers and in-depth interview with aircrews of two randomly selected repatriation flights operated by Thai Airways International: TG476 (Sydney-Bangkok; 209 passengers (female 61.24%; adult 92.82%), 3 pilots, and 14 cabin attendants) on April 26 and TG492 (Auckland-Bangkok; 168 passengers (female 51.79%; adult 51.19%), 4 pilots, and 14 cabin attendants) on April 27, 2020. The Boeing 777 equipped with 18 seats in the business class and 306 seats in the economy class were used for both flights (Fig 1).

Passengers were asked to estimate the distance to the nearest individual(s) at check-in, boarding, and inflight as well as their mobility during the flight. Their opinions and willingness-to-pay for six personal in-flight amenities—disposable food containers, bottled water, gloves, tissue paper, mask, face shield—were assessed using a five-point Likert scale (1, strongly

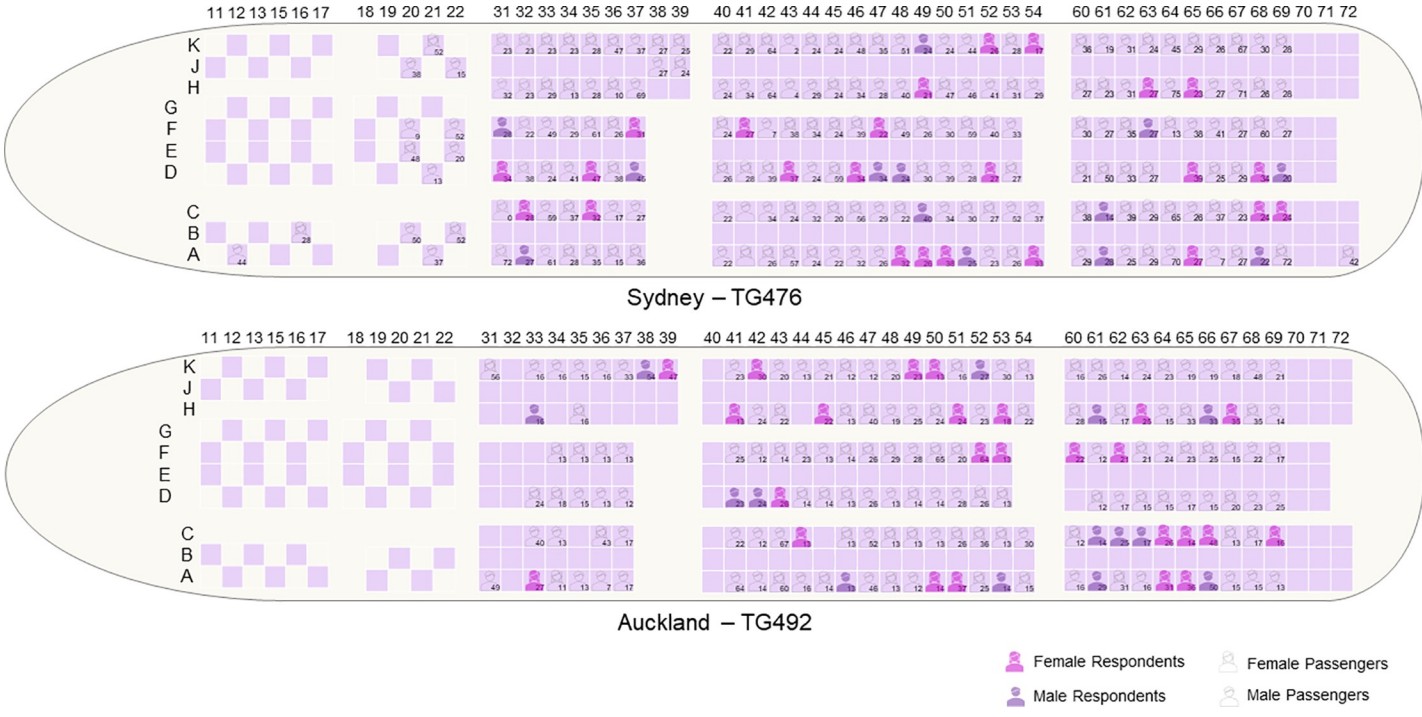

**Fig 1. Passenger seats map of TG476 and TG492 repatriation flights.**

disagree to 5, strongly agree) and an open-ended question, respectively. Passenger's confidence in the company before and after the trip was assessed by using a ten-point scale (1, lowest to 10, highest).

Descriptive statistics (frequency, mean, and standard deviation) were used for data analysis. The response rate was calculated by using responses from passengers at least 18 years of age. Association between categorical variables was analyzed with the chi-square test. Student's t-test was used to compare interval data between groups as appropriate. Likert scale findings were presented as mean and standard deviation for simplicity but the comparison between groups was performed by using chi-square or Fisher's exact test where appropriate.

## Ethics committee approval

This study was approved by the Institutional Review Board of Dhurakij Pundit University.

## Patient and public involvement

The inception of this study was from the discussion with the pilots and cabin crews of the Thai Airways International. They agreed with the simple anonymous survey of passengers and in-depth interviews with aircrews.

## Results

Thirty-seven and forty-one passengers of TG476 and TG492 responded to the survey, respectively. The overall response rate was 22.50% with statistically significant differences between the two flights (18.04% vs 32.56%, respectively; p = 0.007). Mean age and gender distribution of respondents and non-respondents were not statistically different (p = 0.6566 and 0.156, respectively).

**Table 2. Characteristics and experience of passengers in two Thai airways repatriation flights.**

| | Overall | TG476 | TG492 | p-value |
|---|---|---|---|---|
| **Route** | | Sydney-Bangkok | Auckland-Bangkok | |
| **Flight Distance (kilometers)** | | 7,523 | 9,566 | |
| **Flight Duration (hours)** | | 9:25 | 11:50 | |
| **Response Rates** | | | | |
| - Overall | 78/377 | 37/209 | 41/168 | 0.1100 |
| - Age >= 18 | 63/280 | 35/194 | 28/86 | 0.0070 |
| **Age[a]** | 28.14±13.94 | 32.69±13.65 | 22.53±12.17 | <0.0001 |
| **Female** | 57.03% | 61.24% | 51.79% | 0.0650 |
| **Student** | 50.93% | 34.93% | 70.83% | <0.0001 |
| **Flying Experience (times in 2019)[a]** | 3.79±6.07 | 2.64±2.54 | 4.80±7.89 | 0.1192 |
| **Body Temperature (degree Celsius)[a]** | 36.52±0.34 | 36.61±0.34 | 36.40±0.30 | <0.0001 |
| **Physical Distance (meters)[a]** | | | | |
| - Check-in | 1.59±0.48 | 1.57±0.36 | 1.61±0.58 | 0.7020 |
| - Boarding | 1.41±0.52 | 1.42±0.28 | 1.40±0.68 | 0.8507 |
| - In-flight | 1.26±0.65 | 1.27±0.28 | 1.26±0.86 | 0.9239 |
| **In-flight Body Temperature Checked (times)[a]** | 1.97±0.91 | 1.32±0.53 | 2.56±0.78 | <0.0001 |
| **In-flight Mobility (times)[a]** | | | | |
| - Move Around | 2.00±1.65 | 1.54±1.41 | 2.41±1.76 | 0.0186 |
| - To Toilet | 2.08±1.54 | 1.57±1.30 | 2.54±1.61 | 0.0049 |
| **In-flight Personal Amenities[a]** | | | | |
| - Disposable Food Container | 4.59±0.78 | 4.54±0.87 | 4.63±0.70 | 0.6000 |
| - Bottled Water | 4.77±0.64 | 4.73±0.77 | 4.80±0.51 | 0.6095 |
| - Gloves | 4.46±1.02 | 4.46±1.10 | 4.46±0.95 | 0.9864 |
| - Tissue Paper | 4.68±0.67 | 4.65±0.75 | 4.71±0.60 | 0.7038 |
| - Mask | 4.74±0.69 | 4.65±0.89 | 4.83±0.44 | 0.2523 |
| - Face Shield | 4.54±0.88 | 4.46±1.04 | 4.61±0.70 | 0.4540 |
| **Willingness-to-Pay for In-flight Personal Amenities (THB)[a]** | | | | |
| - Disposable Food Container | 60.63±82.15 | 48.89±69.95 | 71.22±91.36 | 0.2331 |
| - Bottled Water | 32.71±65.41 | 19.05±20.75 | 45.02±86.72 | 0.0798 |
| - Gloves | 27.42±59.56 | 14.70±18.53 | 38.90±78.96 | 0.0729 |
| - Tissue Paper | 21.42±30.00 | 12.57±13.09 | 29.41±37.95 | 0.0123 |
| - Mask | 34.04±62.59 | 24.03±36.40 | 43.09±78.56 | 0.1810 |
| - Face Shield | 44.04±45.94 | 29.97±40.67 | 56.73±47.19 | 0.0093 |
| **Confidence in Thai Airways[a]** | | | | |
| - Before | 7.64±2.47 | 7.62±2.49 | 7.66±2.48 | |
| - After | 8.10±2.49 | 8.19±2.46 | 8.02±2.54 | |
| - p-value | 0.0001 | 0.0032 | 0.0144 | |

[a]Mean±SD; THB, Thai Baht (US$1 = THB32.45 as of April 26, 2020)

The mean age of passengers was 28.14±13.94 years old and 57.03% were female. TG492 passengers were mostly students and significantly younger than that of TG476 (p<0.0001) although both groups have comparable flying experience (p = 0.1192) (Table 2, Fig 1). The average body temperature was 36.52±0.34 degrees Celsius.

Passengers estimated average physical distances of 1.59±0.48, 1.41±0.52, and 1.26±0.65 meters at check-in, boarding, and in-flight, respectively. The physical distances at all stages were not different between the two flights. Passengers were checked for body temperature

during the flight 1.97±0.91 times on average which is significantly more frequent in longer than shorter flight (p<0.0001). Likewise, the passengers moved around or went to the toilet during the flight 2.00±1.65 and 2.08±1.54 times which are significantly more frequent in longer than shorter flight (p = 0.0186 and 0.0049, respectively). The passengers agreed with the importance of in-flight personal amenities but were willing to pay for them at varying prices. The confidence in the airline company was statistically significantly increased from 7.64±2.47 before the trip to 8.10±2.49 after the trip (p = 0.0001).

The aircrews were satisfied with the protocol and provided several practical suggestions. Despite the return-to-work intention, they had expressed concerns regarding occupational exposure of themselves and their family members. These concerns seemed to be alleviated after the actual experiences working in the repatriation flights. Physical distancing at approximately 1.5 to 2.0 meters was more practical at the check-in counter, pre-boarding area, and boarding line than during the flight.

The cabin areas were divided by disposable curtains into five designated areas. 'Clean area' was located at the frontmost of the plane, in which only crews with PPE were allowed. 'Buffer zone' was assigned as a dressing area for crews. In the 'passenger sitting area', the initial CAAT requirement to set at least one meter between any two passengers was not feasible for the present seating layout so the repatriation flights asked and received permission from CAAT so that any adjacent seat is empty except for the declared family members. This was also done in the 'quarantine area' (the last three rows), which was for either passengers or crews with unanticipated symptoms just identified onboard. In that case, one cabin crew with PPE will be assigned for the service in the quarantine area and could not be close to the other crews within two meters. 'Lavatories' at the front of the plane were allowed only for crews. Magazines, newspapers, and unnecessary documents were removed.

Cabin crews got dressed in personal protective equipment (PPE) in the buffer zone with no difficulty. However, they reported several occasions in which the crews with PPE crossed paths with the less protected crews. Passengers received surgical mask and face-shield and cleaned their hands with alcohol gel before boarding; however, this approach was not practical for several passengers who had many carry-ons. Before providing the in-flight services, the cabin crews and passengers had to stay only in the assigned zones and minimize their movements. Prepackaged food in disposable containers, utensils, and bottled water were given to individual passengers. The food service was provided at different times, if possible, to minimize the chance of simultaneous mask removal by nearby passengers. Passengers were asked to use the provided alcohol gel to clean their hands before and after the meal. The passengers were asked to drop the garbage to the garbage cart by themselves or on the service tray to minimize physical contact with the cabin crew. The lavatory was disinfected every use.

During the landing, the cabin crews announced that the passengers remain seated and keep the physical distancing while disembarkation. After landing, the cabin crews noticed several passengers attempted to move out too early which might fail the physical distancing principle, so they decided to allow the passengers to stand and disembark on a row-by-row basis. Aircrews moved to buffer zone and take off the PPE before the cleaning staff moved in for aircraft disinfection. All passengers in both flights tested reverse-transcription—polymerase-chain-reaction (RT-PCR) for COVID-19 and were quarantined at a government-provided hotel in Bangkok for 14 days.

## Discussion

The aviation industry has been greatly affected by the COVID-19 pandemic. Several preventive measures have been proposed [5] and some were implemented but might not ensure the

optimal balance between public health risk minimization and airline financial viability. While the diagnostics industry has advanced laboratories and healthcare industry has hospital facilities for producing scientifically robust evidence, the aviation industry has a unique and dynamic context that might not be appropriate for evidence generation. The repatriation flights that received permission to operate during the COVID-19 pandemic have provided a partially controlled setting to collect useful data from passengers' perspective and gather feedbacks from aircrews to assess the implementation feasibility of the mandatory protocol.

The protocol was well received by the passengers and aircrews. Physical distances seemed to be context-sensitive, as suggested by the varying physical distances between the check-in, boarding, and in-flight areas. Estimated physical distance reported by passengers might not be accurate but the data could reflect the subjective perception of passengers which is influential for business.

Some regulations might not have adequate detail so the inputs from the real experiences are essential. For instance, in-flight body temperature check was required for long-haul moderate- and high-risk flights but no frequency was specified. Passengers of the repatriation flights in this study not only agreed with the temperature check but also remained aware that they were approached for a body temperature check.

This study has some limitations. First, the response rates of this voluntary questionnaire survey were low; however, the responses were from an unbiased seat selection and could be representative of the flights. Second, the self-reported data relied on passengers' perception and might not be accurate to be used as a reference for real practice. Third, the nature of Thai passengers might not be exactly like that of other ethnic origins.

## Summary box

- Several preventive measures for in-flight transmission of the SARS-CoV-2 has relied on past experiences and raised financial and feasibility concerns to the airline industry.

- Evidence on the implementation feasibility of commercial airline infection control protocol, especially from the perspectives of passengers and aircrews, has been lacking.

- Our study suggests that the passengers reported varying degrees of physical distancing at check-in, boarding, and in-flight and that the in-flight body temperature check was possible.

- The Thai Airways protocol was well received by the passengers and aircrews.

## Supporting information

**S1 Data.**
(DTA)

## Acknowledgments

The authors would like to thank Dr. Kornprom Saengaram and Vongsa Laovoravit for their kind advice.

## Author Contributions

**Conceptualization:** Krit Pongpirul, Kanitha Kaewpoungngam, Korn Chotirosniramit, Sinnop Theprugsa.

**Data curation:** Kanitha Kaewpoungngam, Korn Chotirosniramit, Sinnop Theprugsa.

**Formal analysis:** Krit Pongpirul, Kanitha Kaewpoungngam.

**Investigation:** Krit Pongpirul, Kanitha Kaewpoungngam, Korn Chotirosniramit.

**Methodology:** Krit Pongpirul.

**Project administration:** Krit Pongpirul.

**Supervision:** Sinnop Theprugsa.

**Validation:** Kanitha Kaewpoungngam, Korn Chotirosniramit, Sinnop Theprugsa.

**Visualization:** Korn Chotirosniramit.

**Writing – original draft:** Krit Pongpirul, Korn Chotirosniramit, Sinnop Theprugsa.

**Writing – review & editing:** Krit Pongpirul, Kanitha Kaewpoungngam, Korn Chotirosniramit, Sinnop Theprugsa.

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
