## [Decision Letter · Decision Letter 0]

16 Jul 2020

PONE-D-20-15480

Commercial Airline Protocol during COVID-19 Pandemic: An Experience of Thai Airways International

PLOS ONE

Dear Dr. Pongpirul,

Thank you very much for submitting your manuscript "Commercial Airline Protocol during COVID-19 Pandemic: An Experience of Thai Airways International" (#PONE-D-20-15480) for review by PLOS ONE. As with all papers submitted to the journal, your manuscript was fully evaluated by academic editor (myself) and by independent peer reviewers. The reviewers appreciated the attention to an important health topic, but they raised substantial concerns about the paper that must be addressed before this manuscript can be accurately assessed for meeting the PLOS ONE criteria. Therefore, if you feel these issues can be adequately addressed, we invite you to submit a revised version of the manuscript that addresses the points raised during the review process. We can’t, of course, promise publication at that time.

We look forward to receiving your revised manuscript.

Kind regards,

Abdallah M. Samy, PhD

Academic Editor

PLOS ONE

**Journal Requirements:**

3. Thank you for including your competing interests statement; "The authors have declared that no competing interests exist."

We note that one or more of the authors are employed by a commercial company:

Thai Airways International Public Company Limited

**Reviewers' comments:**

Reviewer's Responses to Questions

**Comments to the Author**

1. Is the manuscript technically sound, and do the data support the conclusions?

Reviewer #1: Partly

2. Has the statistical analysis been performed appropriately and rigorously? 

Reviewer #1: N/A

3. Have the authors made all data underlying the findings in their manuscript fully available?

Reviewer #1: Yes

4. Is the manuscript presented in an intelligible fashion and written in standard English?

Reviewer #1: Yes

5. Review Comments to the Author

Reviewer #1: 1.This article by Krit Pongpirul gives a research article on commercial airline protocol during COVID-19 pandemic from Thai airways international to explore the implementation feasibility of airways protocol. They conducted an online questionnaire survey of passengers and in-depth interview with aircrews of two flights and they thought that the protocol was well received by the passengers and aircrews of the repatriation flights. Overall, this article gives a broad interview of the potential flight-level prevent and control transmission of SARS-CoV-2 in flght, but lacks depth.

2.The satisfaction of passengers and crew members should not be taken as the main evaluation parameters of the study. The number of research flights should be increased to improve the reliability of the data.

3. 124-125 “The overall response rate of this research was 22.50.” and “Passengers estimated average physical distances” were the results bias?

4. The prolonged flight time is bound to increase the number of temperature tests and the number of activities of passengers on the plane, including eating, going to the toilet, etc. If there are confirmed patients on the flight but they are not detected before boarding the aircraft and subject them to the necessary isolation and treatment, these activities must increase the risk of normal passenger infection.

6. PLOS authors have the option to publish the peer review history of their article (what does this mean?). If published, this will include your full peer review and any attached files.

Reviewer #1: No

---

## [Author Response · Author response to Decision Letter 0]

17 Jul 2020

Point-by-Point responses to Editor’s and Reviewer’s Comments:

Editor: 1. Please ensure that your manuscript meets PLOS ONE's style requirements.

Response: The manuscript was reformatted to meet PLOS ONE’s style requirements.

Editor: 2. We note that you have indicated that data from this study are available upon request. PLOS only allows data to be available upon request if there are legal or ethical restrictions on sharing data publicly. In your revised cover letter, please address the following prompts: a) If there are ethical or legal restrictions on sharing a de-identified data set, please explain them in detail (e.g., data contain potentially sensitive information, data are owned by a third-party organization, etc.) and who has imposed them (e.g., an ethics committee). Please also provide contact information for a data access committee, ethics committee, or other institutional body to which data requests may be sent. b) If there are no restrictions, please upload the minimal anonymized data set necessary to replicate your study findings as either Supporting Information files or to a stable, public repository and provide us with the relevant URLs, DOIs, or accession numbers. For a list of acceptable repositories, please see http://journals.plos.org/plosone/s/data-availability#loc-recommended-repositories. We will update your Data Availability statement on your behalf to reflect the information you provide.

Response: The data has been deidentified and deposited in the UK Data Service. The data was also uploaded to the manuscript submission system while the DOI link is not yet available.

Editor: 3. Thank you for including your competing interests statement; "The authors have declared that no competing interests exist." We note that one or more of the authors are employed by a commercial company: Thai Airways International Public Company Limited. Please provide an amended Funding Statement declaring this commercial affiliation, as well as a statement regarding the Role of Funders in your study. If the funding organization did not play a role in the study design, data collection and analysis, decision to publish, or preparation of the manuscript and only provided financial support in the form of authors' salaries and/or research materials, please review your statements relating to the author contributions, and ensure you have specifically and accurately indicated the role(s) that these authors had in your study. You can update author roles in the Author Contributions section of the online submission form. Please also include the following statement within your amended Funding Statement. “The funder provided support in the form of salaries for authors [insert relevant initials], but did not have any additional role in the study design, data collection and analysis, decision to publish, or preparation of the manuscript. The specific roles of these authors are articulated in the ‘author contributions’ section.” If your commercial affiliation did play a role in your study, please state and explain this role within your updated Funding Statement. Please also provide an updated Competing Interests Statement declaring this commercial affiliation along with any other relevant declarations relating to employment, consultancy, patents, products in development, or marketed products, etc. Within your Competing Interests Statement, please confirm that this commercial affiliation does not alter your adherence to all PLOS ONE policies on sharing data and materials by including the following statement: "This does not alter our adherence to PLOS ONE policies on sharing data and materials.” (as detailed online in our guide for authors http://journals.plos.org/plosone/s/competing-interests). If this adherence statement is not accurate and there are restrictions on sharing of data and/or materials, please state these. Please note that we cannot proceed with consideration of your article until this information has been declared. Please include both an updated Funding Statement and Competing Interests Statement in your cover letter. We will change the online submission form on your behalf. Please know it is PLOS ONE policy for corresponding authors to declare, on behalf of all authors, all potential competing interests for the purposes of transparency. PLOS defines a competing interest as anything that interferes with, or could reasonably be perceived as interfering with, the full and objective presentation, peer review, editorial decision-making, or publication of research or non-research articles submitted to one of the journals. Competing interests can be financial or non-financial, professional, or personal. Competing interests can arise in relationship to an organization or another person. Please follow this link to our website for more details on competing interests: http://journals.plos.org/plosone/s/competing-interests

Response: The Funding Statement was updated as “Thai Airways International provided support in the form of salaries for author ST, but did not have any additional role in the study design, data collection and analysis, decision to publish, or preparation of the manuscript. The specific roles of the authors are articulated in the ‘author contributions’ section.” The Competing Interests Stement was updated as “The authors declare that they have no conflicts of interest. The commercial airline affiliation of author ST does not alter our adherence to PLOS ONE policies on sharing data and materials.”

Reviewer #1: 1.This article by Krit Pongpirul gives a research article on commercial airline protocol during COVID-19 pandemic from Thai airways international to explore the implementation feasibility of airways protocol. They conducted an online questionnaire survey of passengers and in-depth interview with aircrews of two flights and they thought that the protocol was well received by the passengers and aircrews of the repatriation flights. Overall, this article gives a broad interview of the potential flight-level prevent and control transmission of SARS-CoV-2 in flght, but lacks depth.

Response: Thank you very much for the comments. Although we wish we could conduct a more in-depth study, there had been several situational limitations that prevented us from doing so. We decided to conduct the study as best as possible.

Reviewer #1: 2.The satisfaction of passengers and crew members should not be taken as the main evaluation parameters of the study. The number of research flights should be increased to improve the reliability of the data.

Response: The ‘satisfaction’ of passengers and crew members was not the primary focus of our study. Rather, they were ask to report useful information from their perspectives as shown in Table 2. We wish we could increased the number of research flights as advised.

Reviewer #1: 3. 124-125 “The overall response rate of this research was 22.50.” and “Passengers estimated average physical distances” were the results bias?

Response: Despite the low response rate, given the situational limitations mentioned above, we believe that the selection bias was only a minor concern in our study. As mentioned in the Discussion section, the responses were from an unbiased seat selection so the findings could be representative of the flights.

Reviewer #1: 4. The prolonged flight time is bound to increase the number of temperature tests and the number of activities of passengers on the plane, including eating, going to the toilet, etc. If there are confirmed patients on the flight but they are not detected before boarding the aircraft and subject them to the necessary isolation and treatment, these activities must increase the risk of normal passenger infection.

Response: Thank you very much. We agree with the point raised but our data could not be used for testing the hypothesis.

---

## [Editor Report · Decision Letter 1]

27 Jul 2020

Commercial airline protocol during COVID-19 pandemic: An experience of Thai Airways International

PONE-D-20-15480R1

Dear Dr. Pongpirul,

We’re pleased to inform you that your manuscript has been judged scientifically suitable for publication and will be formally accepted for publication once it meets all outstanding technical requirements.

Kind regards,

Abdallah M. Samy, PhD

Academic Editor

PLOS ONE

---

## [Editor Report · Acceptance letter]

29 Jul 2020

PONE-D-20-15480R1 

Commercial airline protocol during COVID-19 pandemic: An experience of Thai Airways International 

Dear Dr. Pongpirul:

I'm pleased to inform you that your manuscript has been deemed suitable for publication in PLOS ONE. Congratulations! Your manuscript is now with our production department. 

Kind regards, 

on behalf of

Dr. Abdallah M. Samy 

Academic Editor

PLOS ONE